# AMOS: A Large-Scale Abdominal Multi-Organ Benchmark for Versatile Medical Image Segmentation

**Yuanfeng Ji**[1], **Haotian Bai**[2], **Chongjian Ge**[1], **Jie Yang**[2], **Ye Zhu**[2], **Ruimao Zhang**[2*]
**Zhen Li**[2*], **Lingyan Zhang**[3], **Wanling Ma**[4], **Xiang Wan**[2], **Ping Luo**[1*]

[1] The University of Hong Kong
[2] Shenzhen Research Institute of Big Data, The Chinese University of Hong Kong (Shenzhen)
[3] Longgang Central Hospital of Shenzhen, China
[4] Longgang People Hospital of Shenzhen, China

## Abstract

Despite the considerable progress in automatic abdominal multi-organ segmentation from CT/MRI scans in recent years, a comprehensive evaluation of the models' capabilities is hampered by the lack of a large-scale benchmark from diverse clinical scenarios. Constraint by the high cost of collecting and labeling 3D medical data, most of the deep learning models to date are driven by datasets with a limited number of organs of interest or samples, which still limits the power of modern deep models and makes it difficult to provide a fully comprehensive and fair estimate of various methods. To mitigate the limitations, we present AMOS, a large-scale, diverse, clinical dataset for abdominal organ segmentation. AMOS provides 500 CT and 100 MRI scans collected from multi-center, multi-vendor, multi-modality, multi-phase, multi-disease patients, each with voxel-level annotations of 15 abdominal organs, providing challenging examples and test-bed for studying robust segmentation algorithms under diverse targets and scenarios. We further benchmark several state-of-the-art medical segmentation models to evaluate the status of the existing methods on this new challenging dataset. We have made our datasets, benchmark servers, and baselines publicly available, and hope to inspire future research. Information can be found at https://jiyuanfeng.github.io/AMOS/.

## 1  Introduction

A dense and pixel-precise understanding of abdominal anatomy is of fundamental importance for computer-aided clinical applications such as disease diagnosis and radiotherapy planning. Specifically, accurate abdominal organ segmentation provides crucial information such as the interrelations among the organs as well as individual positions and shapes in the standardized space, which is essential for assisting clinical decision-making. With the development of related datasets [1, 11, 12, 5, 15], significant progress has been made in this area.

Nevertheless, robust segmentation of abdominal organs remains challenging in a real-world clinical setting because of the large variety of organ morphological structures, appearance, and imaging qualities between images acquired from different patients by different scanners. Some challenges are rooted in the gap between the recent benchmarks and the real-world clinic, which we summarized as follow: (a) *Small-Scale*: Since the acquisition and annotation of 3D Medical data are incredibly expensive, currently available benchmarks contain only a limited number of data samples or organ annotations, or both. (b) *Lack of Diversity*: The tremendous cost also restricts previous works to only acquiring data samples from a single-center, phase, scanner, and disease [1]. Models trained with

---

*Corresponding authors: pluo@cs.hku.hk & ruimao.zhang@ieee.org & lizhen@cuhk.edu.cn

Figure 1: Example annotated slices from `AMOS` dataset. Watch the animations by **clicking** them (Not all PDF readers support playing animations. Best viewed with Acrobat/Foxit Reader). The top and bottom two rows show the CT and MRI slices acquired from different scanners, respectively.

homogeneous and inadequate datasets tend to perform catastrophically when the test data distribution shifts under different clinical scenarios. To bridge the gap, one more reality-oriented and robust solution is urgently needed to be proposed.

To mitigate the above two challenging limitations, we propose `AMOS`, a comprehensive abdominal organ segmentation dataset with abundant annotations of multi-modality, multi-center, multi-scanner, multi-phase, and multi-disease patients, that covers overall 15 organs. Compared with previous benchmarks, `AMOS` owns its unique contributions to the research community from the following aspects, (a) Large-Scale: `AMOS` contains 600 Computerized Tomography (CT) / Magnetic Resonance Imaging (MRI) scans with over 74K annotate slices, which is 20× larger than the previous commonly-used BTCV [12] dataset (3.6K). (b) Diverse and Clinical: `AMOS` acquires data from the real-world clinical settings, where the patients with different abdominal cancers/abnormalities are tested from eight different CT or MRI scanners at two medical centers. The significant heterogeneity poses higher requirements for algorithm robustness. (c) Versatile: Despite the multi-organ segmentation, we also show that `AMOS` could potentially serve as a multi-functional dataset for various learning tasks, such as Out-of-Distribution (OOD) generalization, cross-modality learning, transfer learning, privacy-preserving computation and so on.

With the proposed dataset, we build a new benchmark including existing popular medical segmentation methods. The results prove that current state-of-the-art algorithms fail to make satisfactory performance. Considering the diverse and reality-oriented characteristics owned by `AMOS`, we hope that `AMOS` could serve as a new benchmark to evaluate multi-organ segmentation algorithms in practical applications. To be more specific, the **contributions** of our work to the medial segmentation community are as follow:

- We build a new large-scale, diverse, and clinical abdominal organ segmentation dataset of 600 CT/MRI scans, namely `AMOS`, which is comprehensive with 15 organs. To our knowledge, it is the largest dataset of its kind.

- We benchmark current baseline methods on this newly built dataset with various evaluation metrics, showing the limitation of existing state of the arts abdominal organ segmentation algorithms.

- We carefully design extended experiments to validate that `AMOS` could serve as a versatile dataset for multiple learning tasks.

## 2 Related Work

**Abdominal organ segmentation datasets**    As summarized in Table 1, existing abdominal organs segmentation datasets vary in size, modality as well as the number of annotations. For example, for

| Abdomen Organ Dataset | #Organs | #Scans | #Slices | # Anns per Scan | Modality | Region | Year |
|---|---|---|---|---|---|---|---|
| MSD-Liver[†] [1] | 1 | 201 | 29,402 | 3410K | CT | France | 2018 |
| MSD-Spleen[†] [1] | 1 | 61 | 1,563 | 40K | CT | United States | 2018 |
| MSD-Prostate[†] [1] | 1 | 48 | 712 | 15K | MRI | Netherlands | 2018 |
| MSD-Pancreas[†] [1] | 1 | 420 | 13,141 | 144K | CT | United States | 2018 |
| Kits[†] [8] | 1 | 300 | 23,337 | 997K | CT | United States | 2019 |
| BTCV[†] [12] | 13 | 50 | 3,629 | 431K | CT | United States | 2015 |
| Chaos[†] [11] | 4 | 80 | 1,989 | 52K | CT&MRI | Turkey | 2019 |
| DenseVNet [5] | 8 | 90 | 9,246 | 1430K | CT | United States | 2018 |
| AbdomentCT-1K[†] [15] | 4 | 1,112 | 34,497 | 3412K | CT | Worldwide | 2021 |
| Word [14] | 16 | 150 | - | - | CT | China | 2021 |
| AMOS (ours) | 15 | 600 | 74,026 | 9952K | CT&MRI | China | 2022 |

Table 1: Comparison of AMOS dataset with other conventional abdominal organ segmentation datasets. [†] indicates that the value of the dataset is the estimate from released partial data. - means unavailable estimation due to data inaccessibility.

the single-organ segmentation dataset, the MSD [1], considered as the most commonly-used one, provides annotations of the single organ individually (e.g., the liver, spleen, prostat, and pancreas), with data sizes ranging from tens to hundreds. For the multi-organ segmentation dataset, BTCV [12] takes the pioneering step to provide 50 CT scans covering 13 types of organ annotations. Besides, DenseVNet [5] improves the BTCV dataset via involving additional data, forming a total dataset consisting of 90 scans with eight organ annotations. Compared to the single-modality dataset, the Chaos Dataset [11] provides multi-modality information, including both 20 CT scans for the liver and 20 MRI data with four organs. More recent works aim to include more samples in the specific dataset, e.g., AbdomenCT-1K [15] provides 1,112 scans with four organs, while Word [14] contains 150 CT samples to cover 16 kinds of abdominal organs. Unlike the above works, our AMOS contains 600 CT/MRI scans with 15 types of organ annotation, making it the most comprehensive and diverse benchmark of its kind to date. Besides, considering the underrepresentation problem in clinical datasets, our proposed AMOS from the Asian sub-population will be a valuable resource to the existing pool. We refer the readers to Appendix A for more detailed information on the mentioned datasets.

**Methods for single abdominal organ segmentation** The single organ segmentation has been the dominant task for decades, where numerous solutions have been developed [13, 9, 30, 21, 4, 10, 25]. For example, to precisely segment the liver and tumor, H-DenseUNet [13] designs a hybrid 2D/3D network for better features extraction. By proposing a self-configured framework based on the naive UNet [19], nnUNet [9] achieves superior performance on segmenting the liver, spleen, kidney, as well as pancreas, the approach can be easily adapted to multi-organ segmentation tasks. Besides, to resolve the challenges of small target organs, a series of works [30, 21, 4] adopt the cascaded structures, where the networks are designed in a coarse-to-fine manner. More recent works [10, 25] replace the manual designs by the neural network searching for optimal segmentation architectures to achieve better performance with fewer parameters.

**Methods for Abdominal multi-organ segmentation** For multi-organ segmentation, where multiple organs are segmented simultaneously, networks should be designed to own more powerful ability of discriminating the pixel-wise features. OAN [22] designs a fusion network that takes 2D multi-view images as input and reconstructs the 3D segmentation result finally. DenseVNet [5] proposes a dense 3D network for performance improvement. Besides, due to the data insufficiency, a series of works [29, 3, 27] propose different training paradigms and achieve multi-organs segmentation using partially label annotations from single organ datasets. In this group of work [28, 6, 7, 24], the features patch or image patch are treated as tokens, which are used to conduct efficient non-local context modeling among arbitrary positions, making them achieve SOTA performance on the most popular dataset [12]. However, we empirically found that developing and validating methods with limited data still hinders the potential power of modern deep models, leading to unfair/inaccurate comparisons and estimates of different methods.

## 3 AMOS

AMOS collects both the CT and MRI image data from anonymous patients in clinical medical centers, with abundant segmentation annotations. In this section, we will detail the data curation process for

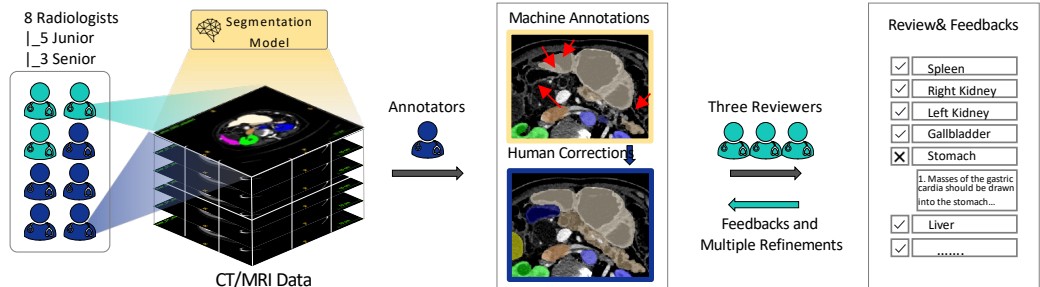

Figure 2: Annotation workflow of AMOS. The coarse annotations automatically labeled by pre-trained segmentors will be further refined by human annotators for multiple times, including 5 junior radiologists for the initial stage and 3 senior specialists for the second checking stage.

both the images and annotations, as well as provide recommendations for potential usages. Additional details, including acquisition, annotation and so on, can be found in Appendix B.

## 3.1 Dataset Construction

**Data Overview**   Following the standard clinical acquisition protocols, the CT/MRI data are collected from 600 patients, who are diagnosed with abdominal tumors/abnormalities at Longgang District People's Hospital and Longgang District People's Hospital, via one of the eight machines as shown in Table 2. Both scanner-generated DICOM and diagnosis reports are collected, de-identified, and stored securely. All data contributions to this study have been approved by the Research Ethics Committees of Longgang District People's Hospital (reference number: 2021077) and Longgang District Central Hospital (reference number: 2021ECJ012).

**Data Collection**   AMOS is designed to facilitate abdominal multi-organ segmentation in a more diverse, clinical, and complex scenarios. To meet this purpose, we cautiously select data, generated by two institutes from 2018 to 2021, based on the following criteria: 1) Patients should be diagnosed with abdominal tumors/abnormalities, while the ones with normal abdomen will be excluded. 2) The imaging quality of the scanned data should be high-quality enough for the radiologists to review and annotate. 3) To ensure the data diversity, the collected samples should be derived from different scanners, as well as different scanning stages. 4) The scan data should cover as many of the specified abdominal organs as possible. To this end, we collected a total of 500 CT and 100 MRI scans from 600 unique patients, covering 15 organ categories, including spleen, right kidney, left kidney, gallbladder, esophagus, liver, stomach, aorta, inferior vena cava, pancreas, right adrenal gland, left adrenal gland, duodenum, bladder, prostate/uterus.

**Data Annotation**   Considering the expensive cost of labeling 3D data, we follow [15, 23] to propose a semi-automatic annotation workflow as shown in Figure 2. There are two stages for annotation, including the coarse labeling stage and the refinement stage. Specifically, a few samples (50 CT and 20 MRI scans) are initially annotated with human labors. Then, we conduct the training process with the annotated scans on several representative models, e.g., 3D-UNet [19] and VNet [16]. The pre-trained segmentors are utilized to pre-label the remaining scans automatically and coarsely. To this end, we finish the coarse labeling stage. For the second stage, five well-trained junior radiologists are employed to check and revisit the segmentation results on a case-by-case basis. To further reduce errors/biases, three senior radiologists with more than 10 years of clinical experience are responsible for the final validation. Typically, they will conduct the annotation review, errors revision, and feedback distribution to improve the annotation quality. The overall process will be iterated several times in the second stage for the final consensus on the well-labeled annotations. Such interactive checking reduces the possible bias caused by individual annotators. We illustrate our semi-automatic annotation workflow in Figure 2. It is evident that human refinements and corrections are required to achieve high-quality annotations. Overall, AMOS annotates 500 CT and 100 MRI scans, resulting into two sub-datasets, i.e., AMOS-CT and AMOS-MRI. We refer the readers for more technical details in the Appendix B.2.

| | Scanner | Domain | #Train | #Validate | #Test | # Total |
|---|---|---|---|---|---|---|
| AMOS-CT | Aquilion ONE | A | 82 | 27 | 27 | 136 |
| — | Brilliance16 | B | 68 | 23 | 23 | 114 |
| — | Somatom Force | C | 50 | 50 | 28 | 128 |
| — | Optima CT660 | D | — | — | 64 | 64 |
| — | Optima CT540 | E | — | — | 58 | 58 |
| | | | 200 | 100 | 200 | 500 |
| AMOS-MRI | Ingenia | F | — | — | 29 | 29 |
| — | Prisma | G | 33 | 20 | 11 | 64 |
| — | Signa HDe | H | 7 | — | — | 7 |
| | | | 40 | 20 | 40 | 100 |

Table 2: Partition of the AMOS dataset. In addition to the conventional train-to-training in-distribution setting, we extend the evaluation setting to out-of-distribution data, where performance is also measured on unseen test data (marked as gray cell)

.

**Data Splits** Distribution shift, where the training distribution differs from the test distribution, is ubiquitous in medical applications. In practice, it is often the case that the testing data (a.k.a., target domain) can differ from training data (a.k.a., source domain) in a variety of ways, such as imaging protocol, device vendors, and patient populations. Such a domain shift always significantly degrade the performance of the developed model. In AMOS, we consider the domain shift caused by the device vendors, where domains are the image acquisition scanner. Besides, the model needs to generalize to data from a new scanner that is disjoint with the training set. Specifically, as shown in Table 2, we make data split according to their performed scanner. For the AMOS-CT, 378 unique abdominal scans from three scanners are randomly split into 200 (n=82, domain A; n=68, domain B; n=50, domain C), 100 (n=27, domain A; n=23, domain B; n=50, domain C), 78 scans (n=27, domain A; n=23, domain B; n=28 domain C) for training, validation, testing (In Distribution, ID) respectively, the other 122 cases (n=64, domain D; n=58, domain E) from other two scanners serve as unseen test data and consist to the testing (Out of Distribution, OOD) split. The same split strategy is adopted to AMOS-MRI, resulting in 40 scans for training, 20 scans for validation and 29 scans for testing (OOD), and 11 scans for testing (ID).

**Data Distribution** All the scanned samples and segmentation annotations are distributed in multiple formats (e.g., DICOM, NIFTI) to ensure their applicability in both the research community and clinical scenarios. Patient-protected health information (PHI) metadata was removed from DICOM files. For additional details on distribution and maintenance, please refer to Appendix B.3.

## 3.2 Dataset Statistics

**Cohort Statistics** To analyze the cohort characteristics of AMOS, we summarize the diagnoses of the collected patients from two aspects (i.e., affected organ and disease type) based on the patient reports. The word frequency results, in Figure 3(a), show that AMOS spreads over a large range of diseases and organs, validating the diversity and variability of our dataset. Due to the page limit, more cohort characteristics (e.g., sexual and age information) are available in Appendix B.4.

**Annotation Statistics** We also conduct the statistical measurements to analyze the semantic label distribution in AMOS. As shown in Figure 3(b), the annotation of AMOS naturally has a long-tail distribution. For example, the scale of liver annotation is about $200\times$ larger than the adrenal gland. This natural/reality-oriented long-tail distribution property makes AMOS more challenging for precise multi-organ segmentation.

**Dataset Comparison** In this part, we compare the data statistics of our dataset to those of commonly used abdominal organ segmentation datasets. Since not all annotations of these datasets are available to the public, some statistics marked with $^\dagger$ are estimated using a subset of the dataset. The overall comparisons are conducted from the aspects of data scale and diversity. First, we compare the data scale between AMOS and other representative segmentation datasets in Table 1. Specifically, AMOS contains 74K annotated slices, which is $2.2\times$ larger than the second-largest dataset AdbomentCT-1K, and $20\times$ larger than BTCV. For the number of scanned samples, our AMOS is significantly larger than most datasets except for AdbomentCT-1K. Nevertheless, AMOS still takes advantage of organ numbers, annotation numbers in a single scan, and modality numbers over AdbomentCT-1K. Second,

| Target \ Source | BTCV [12] | AMOS | MSD-Spleen [1] | MSD-Liver [1] | MSD-Pancreas [1] | Kits [8] |
|---|---|---|---|---|---|---|
| BTCV | 83.62 | **84.47** | – | – | – | – |
| AMOS | 79.74 | **90.83** | – | – | – | – |
| MSD-Spleen | 91.10 | 94.73 | **95.39** | – | – | – |
| MSD-Liver | 92.32 | 95.40 | – | **96.85** | – | – |
| MSD-Pancreas | 66.32 | 71.76 | – | – | **86.68** | – |
| Kits | 80.21 | 88.33 | – | – | – | **97.47** |

Table 3: The model pretrained on the `AMOS` dataset shows superior transfer performance compared to those on the BTCV and MSD dataset, which implies our superior data capacity and quality.

for annotation diversity comparison, Table 8 in the Appendix B.4 shows that the min/max/median values of slice spacing and size range are [0.8/6.0/5.0] and [40/553/115], respectively, which spreads diversely than other abdominal multi-organ datasets. Besides, in term of the organs volume, we visualize the distributions of Chaos, BTCV, AdbomentCT-1K and `AMOS` in Figure 4 in the Appendix. Results reveal that `AMOS`'s distribution almost covers the others, validating its high diversity. Such a property puts a higher requirement on the model's capacity and perception abilities.

To further validate the data diversity and evaluate the annotation quality, we propose an experimental comparison. Specifically, we first pre-train two UNets based on `AMOS` and its closest dataset, BTCV, respectively. The pre-trained models will be employed to infer on multiple target datasets, e.g., MSD-Spleen, MSD-Liver, MSD-Pancreas, and Kits. Given two datasets for comparison, the performance are evaluated on their mutual classes. As shown in Table 3, The model trained on `AMOS` dataset attains 94.73%, 95.40%, 71.76%, 88.33% mDice scores on the target dataset MSD-Spleen, MSD-Liver, MSD-pancreas, Kits dataset, respectively, which significant outperforms the one trained on the BTCV dataset (91.10%, 92.32%, 66.32%, 80.21%), being slightly worse than MSD-pretrained models (95.39%, 96.85%, 86.68%, 97.47%). Besides, the model trained on `AMOS` achieves 84.47% mDice on the BTCV validation set, surpassing the model trained on BTCV training set, which indicates the annotation quality and data diversity of `AMOS` facilitate the models in learning better representations.

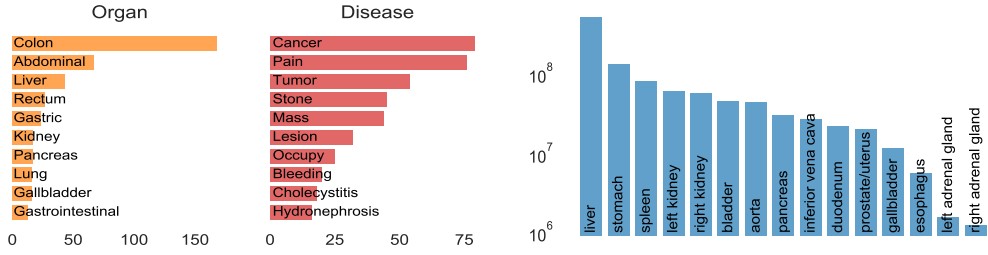

(a) Top-ten most frequent diseases and diseased organs.   (b) Number of annotated voxels per category

Figure 3: Statistics on data targets as well as data annotation, reflecting that AMOS is a clinical, highly diverse data set. The x-axis units of both figures are counts

### 3.3 Evaluation Metrics

Following the previous work [12], we leverage two classical medical segmentation metrics for evaluation, i.e., dice similarity coefficient (DSC), and normalized surface dice (NSD) [17]. DSC score is a popular segmentation metric applied to a variety of segmentation tasks, while the NSD score can provide supplementary information on segmentation quality for evaluating the precision of segmentation boundaries. Specifically, we first calculate the category-wise performances based on the above two metrics and then average the values to obtain the overall evaluation (i.e., mDice and mNSD). For both metrics, a higher score indicates a better result. Besides, considering computational efficiency, we encourage future works to provide information on the model capacity (the number of parameters) and computational cost (GFlops).

### 3.4 Potential Usages

Despite the multi-organ segmentation, `AMOS` provides multi-fold and abundant information, enabling other potential usages in the research community. For instance, in Table 2, we have shown that our `AMOS-CT` and `AMOS-MRI` consist of five and three domains, respectively. It motivates us to conduct corresponding extensions, e.g., OOD generalization, cross-modality learning and transfer learning. Besides, under the general framework of domain shifts, `AMOS` can also be easily adapted to domain

| Model | Params(M) | Flops(G) | CT-Val | | CT-Test | | MRI-Val | | MRI-Test | |
|---|---|---|---|---|---|---|---|---|---|---|
| | | | mDice(%) | mNSD(%) | mDice(%) | mNSD(%) | mDice(%) | mNSD(%) | mDice(%) | mNSD(%) |
| UNet [19] | 31.18 | 680.31 | **88.87** | **79.87** | **89.04** | **78.32** | **85.59** | **80.56** | **67.63** | **59.02** |
| VNet [16] | 45.65 | 849.96 | 81.96 | 67.94 | 82.92 | 67.56 | 83.86 | 78.0 | 65.64 | 57.37 |
| CoTr [24] | 41.87 | 668.15 | 77.13 | 64.15 | 80.86 | 66.31 | 77.5 | 70.1 | 60.49 | 51.18 |
| nnFormer [28] | 150.14 | 425.78 | 85.63 | 74.15 | 85.61 | 72.48 | 80.6 | 74 | 62.92 | 54.06 |
| UNetr [7] | 93.02 | 177.51 | 78.33 | 61.49 | 79.43 | 60.84 | 75.3 | 65.3 | 57.91 | 47.25 |
| Swin-UNetr [6] | 62.83 | 668.15 | 86.37 | 75.32 | 86.32 | 73.83 | 75.7 | 65.8 | 57.5 | 47.04 |

Table 4: Overall results of six baselines method on the `AMOS-CT` and `AMOS-MRI` datasets, respectively. The FLOPS/the number of parameters is estimated by using the $[1 \times 128 \times 128 \times 128]$ patch as model input. The class-wise scores can be found in Appendix C.2.

adaption and sub-population shift problems. Since each domain in `AMOS` can be treated as a separate client, it is also suitable for developing Privacy-Preserving Computation techniques, such as Federated Learning, which are crucial in medical applications. We welcome community contributions towards exploring more potential usages of `AMOS`.

# 4 Experiments

In this section, we perform experimental validation on various datasets to investigate the superior properties of our proposed `AMOS`. First, we introduce the experimental settings, including the selected baseline methods and implementation details. Then, experiments are carefully designed to report the performance from multiple perspectives, with various baseline methods and datasets.

## 4.1 Experimental Settings

**Baseline methods** We select recent state-of-the-art segmentors as baseline methods, including the CNN-based methods (e.g., UNet [19]), Transformer-based methods (e.g., UNetr [7], Swin-UNetr [6]), and the hybrid ones (e.g., Cotr [24], nnFormer [28]). Unless specified otherwise, we follow the default training and testing configurations provided in the published papers or the released codebases to calculate the performances.

**Implementation details** In this work, we use the Pytorch toolkit [18] to conduct all experiments on one single NVIDIA V100 GPU. The nnUNet [9] codebase is adopted for benchmark implementation. For more training and testing details, we refer the readers to Appendix C.1.

## 4.2 Benchmark results

To comprehensively evaluate different segmentors on `AMOS`, we first train representative models on the training split, and then report the corresponding performances in Table 4. The computational costs, i.e., floating-point operations per second (Flops) and the number of parameters, are also reported for efficiency comparisons. Surprisingly, we find that UNet consistently outperforms other recently developed methods for a certain margin, achieving 88.87% and 85.59% mDice, and 79.87% and 80.56% mNSD on `AMOS-CT` and `AMOS-MRI`, respectively. Besides, it's found that the Transformer-based models take no obvious performance advantages over the CNN-based models, with overhead computational consumption. For example, UNet with much less parameters surpasses nnFormer, Swin-UNetr over $2 \sim 3\%$ mDice and $4 \sim 5\%$ mNSD scores on `AMOS-CT`. To further analyze the segmentation performance, we show the detailed category-wise scores in Appendix C.2. Results show that both the Transformer-based methods and CNN-based methods are able to perform well on the large organs (e.g., spleen, liver, and kidney), but still poorly on small organs (e.g., duodenum and adrenal gland). It indicates that the more fine-grained features needed to be captured to assist the segmentors in pixel-wise understanding and precise segmentation. Finally, we have also observed a significant gap between mDice and mNSD as previously mentioned by [15]. Future segmentation algorithms should move toward improving the boundary segmentation accuracy as well as the continuity to achieve better segmentation results.

## 4.3 Generalization Results

To analyze the generalization capabilities of different segmentors, we evaluate their performances on the testing (ID) and testing (OOD) splits, respectively. As shown in Table 6, the performances between two splits vary significantly on `AMOS-MRI`, but almost keep consistent on `AMOS-CT`. For example,

| Model | CT-Test (ID) | | CT-Test (OOD) | | MRI-Test (ID) | | MRI-Test (OOD) | |
|---|---|---|---|---|---|---|---|---|
| | mDice(%) | mNSD(%) | mDice(%) | mNSD (%) | mDice (%) | mNSD (%) | mDice (%) | mNSD (%) |
| UNet [19] | **88.04** | **79.92** | **89.67** | **78.11** | **86.05** | **82.79** | **64.70** | **54.09** |
| VNet [16] | 81.60 | 69.30 | 83.76 | 66.45 | 85.06 | 80.93 | 62.57 | 52.53 |
| CoTr [24] | 79.32 | 67.54 | 81.83 | 65.53 | 78.62 | 71.94 | 57.44 | 46.86 |
| nnFormer [28] | 84.43 | 74.03 | 86.35 | 71.50 | 80.62 | 74.86 | 60.29 | 49.96 |
| UNetr [7] | 77.86 | 62.71 | 80.43 | 59.65 | 74.36 | 65.33 | 55.44 | 43.69 |
| Swin-UNetr [6] | 84.91 | 74.96 | 87.20 | 73.10 | 75.89 | 67.77 | 54.35 | 42.59 |

Table 6: The in-distribution (ID) v.s. out-of-distribution (OOD) performance of models trained with empirical risk minimization. The OOD test set are drawn from data distinct from training data as described in Section 3.1, while the ID comparison are draw from the training distribution. AMOS-MRI dataset show significant performance drop due to to distribution shift , with substantially better ID performance than OOD performance. While the AMOS-CT do not show significant performance gap difference between the ID and OOD test set. We analysis the results in Section 4.3.

for AMOS-MRI, UNet achieves an 86.05% mDice on the testing (ID) set, but only 64.07%($-21.25\%$) mDice on the testing (OOD) set. Similarly, the corresponding performance gap for nnFormer is $-17.68\%$ mDice (80.6% *v.s.* 62.92%). The results demonstrate that there is indeed a data distribution shift caused by collecting MRI samples from different scanners that hinders the model from performing consistently on two sub-sets. However, for AMOS-CT, the performance gap between testing ID and OOD set disappears (e.g., 88.87% *v.s.* 89.04% mDice for UNet). It is because the CT samples collected by different scanners in our work are almost indistinguishable from each other since the scans follow the same imaging standards, including the imaging protocols, intensity, and so on. The detailed properties of the collected data, shown in Appendix B.4, also validate the imaging intensity is consistent in AMOS-CT but distinguishable in AMOS-MRI. Due to the different data distributions in AMOS-MRI subsets, we encourage the research community to explore AMOS-MRI for more OOD generalization tasks.

## 4.4 Extended explorations on AMOS

Since AMOS contains more modalities and abundant annotations than other works, we are able to additionally investigate the following properties of AMOS, which haven't been explored by previous abdominal segmentation datasets.

**Cross-Modality learning** Though CT and MRI contain internal body information, the information that they are good at capturing is different. For instance, CT shows more potential in imaging organs and skeletal structures, while MRI especially focuses on more fine-grained organ details and soft tissues. An intuitive motivation is that the data from two modalities (i.e., CT

| Training data | CT-Val | MRI-Val |
|---|---|---|
| Individual (200CT / 40MRI) | 88.87/79.97 | 85.59/80.56 |
| Joint (200CT+40MRI) | 89.42/80.78 | 87.73/82.72 |
| Joint (160CT+40MRI) | 89.12/80.19 | –/– |
| Joint (10CT+30MRI) | –/– | 86.28/80.96 |

Table 5: Cross modality learning results. The performance is reported in mDice/mNSD.

and MRI) could complement each other for better clinical diagnosis. Here, we conduct experiments to improve the segmentation performance via cross-modality learning. Specifically, we first train the UNet model on three datasets (AMOS-CT, AMOS-MRI, and their joint), and then evaluate its performance on each modality. Results in Table 5 show that the jointly trained model consistently improves the individually trained model on AMOS-CT (i.e.,+0.55% mDice and +0.81% mNSD) and AMOS-MRI (i.e., +2.14% mDice and +2.16% mNSD). To mitigate the effect that the improvement is caused by more training data, we conduct two comparative experiments by selecting training samples randomly (i.e., 160CT+40MRI, 10CT+30MRI). The improvement could also be observed, which validates the effectiveness of cross-modalities training.

**Transfer Learning** Models trained on the large-scale and diverse dataset are usually supposed to have stronger transfer learning abilities to perform well on the related sub-sets. We conduct experiments in this part to validate how AMOS benefits models in transfer learning scenarios. Taking UNet as the baseline method, we train the models with two training configurations: 1) the model is trained on the specific dataset from scratch, 2) the model is first pre-trained on AMOS and then fine-tuned on the other datasets. The performances are evaluated on 10 segmentation tasks, including six related datasets that contain organs in AMOS, and four unrelated ones. The overall tasks are suitable for evaluating the transfer learning abilities of AMOS, since they contain both the seen and unseen

| | Train from scratch | | Pretrained with AMOS | |
|---|---|---|---|---|
| | mDice(%) | mNSD(%) | mDice(%) | mNSD(%) |
| MSD-Liver [1] | 80.31 | 57.40 | 81.97↑ | 59.14↑ |
| MSD-Spleen [1] | 95.39 | 88.40 | 96.40↑ | 90.12↑ |
| MSD-Pancreas [1] | 66.71 | 47.89 | 66.17 | 48.32 |
| MSD-Prostate [1] | 80.64 | 56.75 | 81.47↑ | 58.73↑ |
| Kits [8] | 88.52 | 81.97 | 89.93↑ | 83.62↑ |
| SegTHOR | 89.66 | 74.77 | 90.91↑ | 75.29↑ |
| MSD-Cardiac [1] | 93.89 | 86.58 | 93.67 | 85.89 |
| MSD-HepaticVesse [1] | 70.21 | 61.5 | 70.36 | 61.79 |
| ACDC [2] | 92.30 | 93.64 | 92.45 | 94.05 |
| Covid19-Seg [20] | 72.53 | 56.22 | 72.68 | 56.51 |

Table 7: Transfer learning results on 10 target datasets.

domains. We detail these datasets and the corresponding experimental details in the Appendix C.2. Results shown in Table 7 demonstrate that model trained on AMOS consistently outperforms the one in train-from-scratch manner on 5/10 benchmarks, while performing comparably on the other 5/10 tasks. It indicates when the sub-sets are related to AMOS, the representations trained by AMOS are transferable and beneficial. On the other hand, when the sub-sets domains are significantly different from AMOS, e.g., MSD-Cardiac, AMOS's representations will not cause the deleterious effects.

## 5 Discussion & Conclusion

In this work, we present a large-scale, diverse, clinical dataset for abdominal multi-organ segmentation, termed AMOS. Thanks to its abundant information, we are able to explicitly focus on measuring algorithm performance fairly and comprehensively. The extensive experiments on AMOS show that our dataset can not only serve as a general multi-organ segmentation benchmark, but also assist the research community in extended applications, including OOD generalization, cross-modality learning, and transfer learning, which haven't been explored by previous works. We believe that our work provides an important step toward the dense, pixel-precise abdominal anatomy understanding and other future explorations. For the limitation, first, all data were acquired from patients who received an abdominal scan at two hospitals, which is a subgroup of the Asian population. Second, although data were collected from different scanners, both scanners were from the two hospitals, which may affect model performance on data acquired from the different medical centers. Additionally, there are other anatomical structures in the scans (e.g., lung, heart), which is not densely annotated but can help medical workflow. In future work, we will look to curate data from a more wide patient pool and multi-center scanners and add annotations for additional organs. Besides, we carefully solve privacy and social effects by strictly obtaining the Research Ethics Committees Approvement.

## Acknowledgement

This work is supported by the General Research Fund of HK No.27208720, No.17212120, and No.17200622, by the Open Research Fund from Shenzhen Research Institute of Big Data No. 2019ORF01005, by the Young Scientists Fund of the National Natural Science Foundation of China under grant No. 62106154, No. 61902335, by Guangdong Province Basic and Applied Basic Research Fund Project Regional Joint Fund-Key Project 2019B1515120039, by the Guangdong Provincial Key Laboratory of Big Data Computing.

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
