# OpenReview forum: "AMOS: A Large-Scale Abdominal Multi-Organ Benchmark for Versatile Medical Image Segmentation"
_NeurIPS.cc/2022/Track/Datasets_and_Benchmarks — NeurIPS 2022 Datasets and Benchmarks _

### Official Review · Reviewer_CcuT · 2022-07-26
**This paper proposes a large-scale abdominal multi-organ dataset under versatile scanning conditions for medical image segmentation. This dataset may overcome the ongoing ground-truth data deficiency problem for training large-scale models on diverse clinical scenarios. Each instance in the dataset is labeled with voxel-level annotations of 15 abdominal organs. Experiments were conducted for cross-modality and transferability of the dataset.**

**Rating:** 7
**Confidence:** 4
**Clarity:** Yes

**Strengths:**

Wide variety in data collection. Labels for a large number of organs. Large size of the dataset. Benchmarking with state-of-the-art methods. Ablative study and dataset exploration.

**Weaknesses:**

More experiments are required. Dataset marginally stands out in comparison to existing datasets with the present set of experiments.

**Additional Feedback:**

None

**Correctness:**

Although many experiments have been conducted on this dataset, however, it is hard to conclude the superiority/hardness of this dataset over other existing ones. In this regard, authors may have shown more results involving other methods than UNet and conducted the same experiments (Table 3, 5, 6, 7) over other different datasets or under the reversed setting.

I believe that this paper will be a valuable contribution to the field, and I recommend its acceptance with some clarifications. Below I have a few questions and comments for the authors.

- How is the second challenge about lack-of-diversity addressed? And is there any effect of these diverse features (of the dataset) on the performance of models? It should be quantified through experiments.
- Table 1 may also have a column indicating the diversity available in the datasets. Here, authors should also mention the class imbalance in the dataset.
- Table 3 presents an interesting transfer learning analysis. Please tell why other methods ([6],[16]) were not used than UNet.
- In Table 1, why are MSD datasets written separately per organ? Isn't it one dataset and the largest dataset?
- In Table 3, why are results on MSD dataset better than AMOS? Why is the performance of models pretrained on MSD dataset not shown for AMOS and BTCV?
- In Table 7, I suspect that what authors call OOD is just the test split and not truly the OOD data since the test and train distributions are similar. This may be the reason behind better performance on OOD data than ID for CT images in Table 7. If that is the case, authors should conduct pure OOD experiments. A more stringent setting could be testing on AbdomentCT-1K, Chaos, Word, or DenseVNet.
- From Table 8, can we conclude that we don't need a very high number of CT images to train the network? As the performance of (10CT/30MRI) is at par with (200CT/40MRI). What do these results indicate about the ground-truth data deficiency problem? Based on results from Table 8, I think it is not very conclusive to say how large of a dataset we need to overcome ground-truth deficiency problem?
- Authors may write a small para about: How feasible is the extension of this dataset? What points need to be considered while extending it?
- Further experiments should be conducted to show transferability results with methods [6] and [16], in Table 6.
- Please check the order of tables. Table 6 should not appear after table 7.

**Documentation:**

Yes, there is sufficient detail to support reproducibility.

**Ethics:**

No ethical concerns for discussion.

**Relation To Prior Work:**

Yes, authors have covered sufficient depth and width of recent literature.

**Summary And Contributions:**

This paper provides a comprehensive dataset for abdominal multi-organ image segmentation. The dataset is supported with statistical analysis and experimentation to evaluate its suitability over existing datasets. The paper is well written and provides updated relevant literature. The authors have given necessary details about dataset construction, data evaluation and splitting. This dataset will help research in the field of abdominal multi-organ segmentation. It adds a variety of images to the existing data. Many state-of-the-art methods are used to benchmark the dataset.

---

> ### Author Response · Authors · 2022-08-24
> **Reply to Reviewer CcuT**
>
> Dear Reviewer CcuT,
>
> Thank you for your comments. We answer the raised questions and concerns below. We have included all discussions and results in our revised manuscript (blue content).
>
>
> **1. Extended experiments for better comparisons**
>
> Thanks for the suggestions. In fact, we have included the most common-used methods for benchmark comparisons. For the future work, we will add more up-to-date methods for the completeness of experiments.
>
> **2. How to address the challenge of lacking diversity**
>
> Thanks for this insightful question. To solve this one, we construct new dataset from multi-center, multi-vendor, multi-modality, multi-phase, multi-disease patients. In addition to this, the diversity of data annotation (multiple organs) alleviates this challenge to some extent. From Table 4 as well as Table6, we can see the performance gains from increasing data diversity.
>
>
> **3. Descriptations of the data diversity and class imbalance**
>
> Thank you for your suggestions. However it is difficult to quantify the diversity of the dataset in practice. Nevertheless, we provide an analysis of the variability of the annotation size in Figure 4, which reflects the diversity of the data to some extent. For class imblance problem, we will improve it in the revision.
>
> **4. Clarifications on the experiments of UNet**
>
> In fact we have consistently achieved the best performance with UNet under large-scale experiments, since the training configurations of UNet have been provided by the previous works. Thus we choosed it as the method for many experiments. Nevertheless, we also agree with you about conducting sufficient experiments. Due to the time limitation, we will add more methods to the conduct the experiments in the revision.
>
> **5. Descriptions on MSD dataset**
>
> Thanks, MSD provides a total of ten sub-datasets, where each contains the annotation of a separate organ, e.g. MSD-Spleen only provides the annotation of the spleen.
>
> **6. Why MSD performs better**
>
> Thanks, the reason why the results on the MSD dataset perform better than AMOS is that there is a shift in the data distribution between AMOS and MSD.
>
> **7. Clarifications on OOD experiments**
>
> Thanks, Thank you for pointing this out. Due to time limitation, we will incorporate more additional possible datasets [a] for OOD testing for the future work.
>
> **8. Necessity of the high number of CT images**
>
> Thank you for pointing it out. There may be a slight misunderstanding that (200CT/40MRI) is the result of training each of the two modalities separately, while (10CT+30MRI) is the effect of testing a total of 40 samples. Considering that (200CT+40MRI achieved 87.73 results on MRI-val, we claim that 1.5mAP is still a big improvement and the amount of data is necessary. For the future work, we will follow your suggestion to see if increasing the amount of CT data can have a boost on top of MRI-OOD, i.e., improving the ability of OOD.
>
> **9. How to extend AMOS**
>
> Thank you very much for your comments. Actually, AMOS can be extended in many ways, such as adding more centers, including more modal data, increasing the annotations types, and annotation of other tasks based on AMOS images, etc.
>
> **10. Extension on methods**
>
> Thank you for your comments. Due to the time limitation, we leave it for the future work to add more up-to-date/popular methods for the completeness of experiments.
>
> **11. Incorrect organizations of tables**
>
> Thank you for the detailed suggestions. We have fixed the table organizations in the latest revision.
>
> [a] Kavur, A. E., Gezer, N. S., Barış, M., Aslan, S., Conze, P. H., Groza, V., ... & Selver, M. A. (2021). CHAOS challenge-combined (CT-MR) healthy abdominal organ segmentation. Medical Image Analysis, 69, 101950.

---

> > ### Comment · Reviewer_CcuT · 2022-08-29
> > **Satisfied with the revision**
> >
> > Thanks for the update, I have no more questions.

---

### Official Review · Reviewer_U7Pt · 2022-07-26
**AMOS: A Large-Scale Abdominal Multi-Organ Benchmark for Versatile Medical Image Segmentation**

**Rating:** 9
**Confidence:** 5

**Strengths:**

To date, this is the largest such database that is being shared with the research community. The project provides a complete annotation workflow for the labeling process. A number of baseline models were developed to evaluate the dataset.


**Weaknesses:**

Labels provided are sparse. There are no sociodemographic features that will enable interrogation for algorithm bias of downstream models. The absence of information on comorbidities is also a limitation of the dataset.

**Additional Feedback:**

Appendix B.5 is missing.

**Clarity:**

Submission is well-written. However, there are a few questions that need further elaboration:
●  	How is the performance in the first stage of annotation using a segmentation model trained with a small number of images?
●  	Was de-identification performed on the images?

**Correctness:**

It is misleading to characterize the database as "diverse" given that sociodemographic labels and comorbidities are missing. But the dataset was meticulously constructed.

**Documentation:**

Documentation is solid.

**Relation To Prior Work:**

Comparison to previous related work on medical image segmentation is comprehensive.

**Summary And Contributions:**

The authors constructed an abdominal database for multi-organ segmentation with 660 CT/MRI exams and 74.000 slides from 8 distinct CT and MRI scanners in two Hong Kong centers.

---

> ### Author Response · Authors · 2022-08-24
> **Reply to Reviewer U7Pt**
>
> Dear Reviewer U7Pt,
>
> Thank you for your comments. We answer the raised questions and concerns below. We have included all discussions and results in our revised manuscript (blue content).
>
> **1. Missing sociodemographic labels and comorbidities**
>
> Thanks for pointing out the limitations on sociodemographic features. We admit including more cases from other populations will enlarge the dataset diversities as well as enable interrogation for algorithm bias of downstream models. For future work, we will enlarge the datasets by includung more cases from more centers, if funding permites.
>
> **2. The performance of automatic annotation**
>
> This is a good question worth discussing. Actually, we have evaluated the overlap between the initial annotations generated by the model and the annotations that were finally reviewed and approved using dice. It turns out that the average dice score is about 67, which indicates that a lot of human effort is still needed to further improve the annotation qualities, even though the model has reduced the workload a lot.
>
> **3. Clarifications on de-identification performed on the images**
>
> Yes, we also check the image case by case make sure not any PHI information on it.
>
> **4.Appendix B.5 is missing**
>
> Sorry for the mistake. We have append the latest appendix in the revised manuscript. Please see the blue content in the updated version.

---

### Official Review · Reviewer_vCQ1 · 2022-07-26
**This work presents a large-scale multi-label abdominal segmentation dataset. While the dataset is useful, some crucial information is missing, which hinders reproducibility and better understanding of this dataset.**

**Rating:** 8
**Confidence:** 4

**Strengths:**

- Scale: the dataset is currently the largest abdominal segmentation dataset available. Additionally, AMOS provides segmentation masks for the most number of organs.
- Diversity: Most abdominal organ segmentation datasets available are curated from hospitals in the US or Europe. AMOS improves the diversity of current datasets by providing images from the Asian population.
- The dataset also includes scanner information (form 8 scanners) and provides an out-of-distribution test set, which allows robustness and out-of-distribution studies.
- The authors provide a suite of strong baselines to benchmark their dataset, including convolution and transformer-based models.
- The Transfer learning experiment (Table 3) future demonstrates the value of AMOS for medical image segmentation.
- All figures and tables are well made. The animated figure (Figure 1) is a nice touch.


**Weaknesses:**

- The manuscript has a truncated Appendix, which left out a lot of crucial implementation and dataset details.
- To better test for robustness and fairness, the dataset should include more metadata such as patient demographics. The accompanying reports can also be really useful for the research community.
- Unclear why only diseased patients are included, especially since no disease label or diseased level segmentations are provided. These disease level labels are tremendously helpful for the medical imaging research community.
- More implementation and data processing details are needed. For instance, CT scans are known to be sensitive to the viewing window level and window width. Such information should be included to ensure reproducibility.
- No Github link is provided. Thus reproducibility is further limited.
- The authors use “default training and testing configurations” for the baseline method. This isn't the optimal approach since neural networks are known to be sensitive to hyperparameters. Thus more extensive hyperparameter search is necessary.
- Table 6 compares models trained from scratch with models pretrained with AMOS. A more comprehensive comparison should include models that are imagenet pretrained.
- The authors write in section 4.3 that “the CT samples collected by different scanners in our work are almost indistinguishable”, and proved their point using results from table 5. However, this contradicts their claim of having a diverse dataset that can be tested or OOD by providing scanner information.


**Additional Feedback:**

N/A

**Clarity:**

Writing
- The paper is organized well. However, typographical and grammatical errors should be corrected.
- Certain wording and sentences are hard to understand without further clarification. For example, L182 - what are slices spacing and size range? Table 4 caption - what is the [1x128x128x128] feature map?


Tables/Figures
- Please bold best performances in all tables
- Table 7 comes before table 6, the labels should be flipped

Dataset checklist:
- Where exactly is the potential negative societal impact provided?
- Paper did not provide codebase

Appendix
- Missing text for A.1
- Missing everything after A.2


**Correctness:**

The data collection is constructed nicely, however, the authors should specify why only diseased patients are included. The evaluation methods are comprehensive but the authors should perform more hyperparameter tuning.

**Documentation:**

The paper is missing some crucial information, largely due to the truncated Appendix. Information on availability and maintenance, and ethical and responsible use is also limited.

**Ethics:**

The authors indicated that “All data contributions to this study have been approved by the Research Ethics Committee” in the main manuscript. However, the authors also put “N/A” for “links to IRB” in the dataset checklist.

**Relation To Prior Work:**

The paper provides a comprehensive review of prior work (summarized in Table 1), and showed improvement in scale and diversity. The authors also included strong baselines from multiple representative segmentation models.

**Summary And Contributions:**

This work presents AMOS, a large-scale dataset that includes segmentation masks for 15 different organs for 500 CT scans and 100 MRIs. In addition to the large scale, AMOS is also relatively diverse, including medical images generated from 8 different scanners from 2 different hospitals in China. Such a heterogeneous dataset can promote algorithmic robustness and out-of-distribution study, which are important topics for medical images.

---

> ### Author Response · Authors · 2022-08-24
> **Reply to Reviewer vCQ1**
>
> Dear reviewer vCQ1,
>
> Thank you for your comments. We answer the raised questions and concerns below. We have included all discussions and results in our revised manuscript (blue content).
>
> **1. Truncated Appendix**
>
> We are sorry for the truncated and uncomplete appendix. We have appended a complete and correct appendix in the latest revision. Please have a further check.
>
> **2. Including more meta data**
>
> Thank you for your valuable comments. In fact, we are working on the mentioned meta-data processing. Specifically, we are converting the corresponding Chinese medical terms into English, although it is time-consuming.
>
>
> **3. Why only include dieseased patient?**
>
> Thanks for this suggestion. Please see **Clarifications on including more normal cases** in the **Author Response to All Reviewers**. We would like to claim that including the normal cases will be truly beneficial to general organ segmentations. However, considering the clinical scenarios in practice, AMOS mainly focuses on the abnormal organs. For future work, the normal cases will be considered to be included for dataset diversities.
>
>
> **4. More implementation and data processing details**
>
> Thanks for the comments. We have included more implementation and data processing details in the latest Appendix B.
>
> **5. The available Github link**
>
> We have updated the repository link in the latest Appendix C. In the future, we will manage all the codes (https://github.com/JiYuanFeng/AMOS2022) in the github platform.
>
> **6. More hyperparameters tuning**
>
> Thanks to point out the weakness. In fact, we did a series of experiments with different hyper-parameters for Swin-Unetr [a]. However, it turned out that we found no significant differences among the experimental results. Thus, we finally adopted the default configuration for simplicity and fair comparisons. Nevertheless, we fully agree with your comments, and we will conduct more thorough experiments in the future to conduct further comparisons.
>
> **7. Explore Imagenet-pretraining for 3D medical segmentation task**
>
> Thanks for the kind suggestions. Actually the 2D pre-trained models usually have a poor performance on the 3D tasks, since the data domains from the pre-trained tasks and the down-stream tasks vary. However, we will conduct the related experiments in the furture work.
>
> **8. No OOD shift on the CT images**
>
> Thank you for pointing it out. In fact, MRI occurs with a much larger offset than CT. Besides, we pointed out in the line 264-266 that using MRI would be a good testbed for resolving OOD. In addition, we apologize for the misunderstanding here. The diversity of AMOS derives from different aspects, including the multiple scanners, the patients acquired by each scanner, and the diverse labeling, as can be seen in Figure 4 in the appendix section.
>
> **9. Some typos and misunderstandinng words**
>
> Thanks for the detailed suggestions. We have fixed all the metioned typos in the latest revision (see the blue content in the main paper). Beside, for the mentioned words that are difficult to understand, we have also added the corresponding clarifications.
>
> **10. Bold best performance on all tables**
>
> Thanks for the suggestions. We have bolded best performance in all tables.
>
> **11.Missing important information**
>
> We have updated the missing information. Please check it in the latest revision.
>
> [a] Hatamizadeh, A., Nath, V., Tang, Y., Yang, D., Roth, H. R., & Xu, D. (2022). Swin unetr: Swin transformers for semantic segmentation of brain tumors in mri images. In International MICCAI Brainlesion Workshop (pp. 272-284). Springer, Cham.

---

> > ### Comment · Reviewer_vCQ1 · 2022-08-28
> > **Satisfied with the revision**
> >
> > Thank you for addressing my questions and comments. I am satisfied with the revision and have updated my rating based on your changes.

---

### Official Review · Reviewer_jTzG · 2022-07-27
**A multi-center, multi-scanner dataset of 600 CT and MRI scans, with labels for 15 organs**

**Rating:** 7
**Confidence:** 4

**Strengths:**

The AMOS dataset has many strengths. Most of all, the dataset is diverse. It includes 5 CT scanners, 3 MRI scanners, and 2 institutions. This is tremendous strength of AMOS, because it can help expose overfitting or lack of generalization in models (the authors emphasize this in the section "Data Splits"). In addition, this is a dataset from Chinese hospitals (that are presumably serving mostly Chinese patients), so this will add much-needed diversity to the field.

**Weaknesses:**

This work overall has few weaknesses in my opinion, but there is one very important weakness: dataset availability and accessibility. The dataset is currently available via the Grand Challenge website, after also submitting a Google form and being accepted as a user. But are there any expectations or guarantees for this the AMOS22 challenge site to remain available to the public? The authors should consider a different way of sharing the data -- one that does not require an account on the Grand Challenge website. In addition, potential users of the data need to submit a form to access the data. Can the authors clarify what would disqualify someone from gaining access to the data? If at all possible, I would highly encourage the authors to consider making their data available on a platform like Zenodo.

Another weakness is that the appendix seems incomplete. Only sections A.1 and A.2 are included (and A.1 is empty). The table of contents of the appendix suggests that it includes important details about the dataset, including about acquisition and annotation.

Last, no code repository is included with this submission. The code to conduct the benchmarking experiments would foster reproducibility. _Without this code, the manuscript is not reproducible._

**Additional Feedback:**

There should be a citation for lines 31-33: "The tremendous cost also restricts previous works to only acquiring data samples from a single-center, phase, scanner, and disease. Models trained with homogeneous and inadequate datasets tend to perform catastrophically when the test data distribution shifts under different clinical scenarios."

In lines 75-77, the authors write "Unlike the above works, our AMOS contains 600 CT/MRI scans with 15 types of organ annotation, making it the most comprehensive and large-scale benchmark of its kind to date." In the previous sentence, however, the authors mention the AbdomenCT-1K dataset, which includes 1,112 scans. I am under the impression that with the phrase "most comprehensive and large-scale benchmark of its kind to date", the authors are implying that their dataset is the largest of its kind to date. Can the authors please rephrase this to make it clear that it is not the largest --- but it does seem to be the most comprehensive in terms of number of organs annotated. And it is indeed a good sized dataset, especially considering the size of related datasets.

The authors reference the U-Net and nnUNet in "Methods for single abdominal organ segmentation." It is true that the U-Net can be used for single-organ segmentation, but it can also be used for multi-organ segmentation. The authors should clarify this in this section.

In the paragraph on "benchmark results", the authors mention that segmentations of small organs are still relatively poor, and the authors highlight a need to improve this. The authors might consider referencing methods that are designed to address segmenting small objects, like the Tversky loss or Focal loss of FocalTversky loss.

Below, I included a few minor comments.

Spelling / word choice comments:
- line 20: "computer-aided diagnosis applications" could probably be "computer-aided clinical applications"
- line 25: "Nerveless" should probably be "Nevertheless" (?)
- lines 93-94: "a bunch of works" seems too casual for a scientific manuscript.
- line 188: should "closed" be "closest"?
- line 205: "better score" should probably be "higher score"

**Clarity:**

Yes, overall the paper is very well written. I have a few specific comments about clarity:
1. I do not understand the phrase on lines 67-68: "with data volumes ranging from tens to hundreds." Please clarify this in the text.
2. The authors frequently reference the "mDice" metric. How does this differ from the Dice coefficient? Please clarify in the text.

**Correctness:**

The dataset appears to be constructed in a sound way, and the benchmark experiments seem appropriate.

**Documentation:**

The authors do a great job of detailing data collection and organization. I have some clarifying questions below.

1. Can the authors please describe how the data were de-identified? I ask the same question in the "Ethics" section of this review. In particular, I wonder about de-identification in the DICOM files.
2. In the "Data Collection" paragraph, the authors describe criteria for inclusion. Was image resolution considered? I would be curious to know the range of 3D resolutions in the included data. If this information is included in the manuscript, I apologize for missing it.
3. Can the authors be more specific regarding the medical training of the junior radiologists? For example, approximately how many years they have been training as a radiologist after completing their initial medical school education. To me, this knowledge is important to help gauge the quality of the segmentations. Thank you for including the number of years of education for the senior radiologists.
4. How were the data converted from DICOM to NIFTI? What software was used?
5. The authors should provide details on ethical and responsible use of this data. This seems related to the "Potential Usages" paragraph.
6. In Figure 3a, I am confused about the diseases listed. Cancer, tumor, and mass are all potentially related (depending on the tumor or mass, they could all be potentially combined as "cancer"). "Occupy" is also not a disease that I know of -- can the authors please clarify? Perhaps my issue with this is that the figure title is "Disease", but many of the labels are not diseases on their own (pain, occupy, bleeding). Can the authors please use a more precise title for this plot?

**Ethics:**

The data are distributed in DICOM format, which can carry metadata about a patient. The authors write that they have de-identified the data. Can the authors please describe in detail how they have de-identified the data?

**Relation To Prior Work:**

Yes, the authors nicely summarize previous work and describe the novelty of their work. However, the paragraph titled "Methods for abdominal multi-organ segmentation" should almost certainly include a reference to the U-Net. In addition, the authors write (lines 98-100) that "However, we empirically found that developing and validating methods with limited data still hinders the potential power of modern deep models, leading to unfair/inaccurate comparisons and estimates of different methods", but it is unclear whether they are referencing this work or a previous work of theirs. Can the authors please clarify?

**Summary And Contributions:**

The authors introduce a dataset for the segmentation of multiple organs in the abdomen, from real-world clinical 3D CT and MRI scans. The authors name their dataset AMOS, and it is meant to address two limitations in previous benchmark datasets: small-scale and lack of diversity. AMOS contains 500 CT images and 100 MRIs, each containing segmentations for 15 organs. The data come from two centers and eight scanners and include several different diseases related to abdominal organs. Importantly, healthy abdominal scans are not included in this dataset. The authors also conduct several experiments that evaluate the dataset in relation to previous datasets and benchmark several segmentation models on the AMOS dataset. This is impressive work that must have required close collaboration between computer scientists and clinicians -- kudos on the great interdisciplinary work!

---

> ### Author Response · Authors · 2022-08-24
> **Reply to Reviewer jTzG**
>
> Dear reviewer jTzG,
>
> Thank you for your comments. We answer the raised questions and concerns below. We have included all discussions and results in our revised manuscript (blue content).
>
> **1. Dataset availability and accessibility**
>
> Thank you for your valuable comments. We are currently building an individual website for AMOS. Besides, as you suggested, we are also going to host the data on the zendo platform. Once the work is done, we will update the latest information to the revision paper.
>
> **2.Missing appendix**
>
> We are sorry for the uncomplete and non-latest appendix in the submmitted version. We have updated the appendix to the latest version to make sure that there’s no missing and uncomplete sub-sections.
>
> **3.Misssing code repository link**
>
> We have updated the repository link in the latest Appendix C. We will manage all the codes (https://github.com/JiYuanFeng/AMOS2022) in the github platform.
>
> **4. Clarifications on the "*data volumes*"**
>
> We are sorry for the misunderstanding caused by the inappropriate term. In fact, it should be ***data size***, which indicates the number of data samples. We have updated the term in the latest revision.
>
> **5. Clarifications on the term "mDice"**
>
> We use mDice to indicate the mean Dice, which is the average of the dice scores of all organ categories. We have added a description of the term in the latest revision.
>
> **6. Clarifications on the process of de-identification**
>
> As described in line 160, Patient-protected health information (PHI) metadata was removed from DICOM files. Besides, other private information is also carefully removed and re-checked in AMOS.
>
> **7. Resolution of the collected data**
>
> Thank you for pointing it out. In fact, we have already introduced a limitation on the resolution in the second point of our data collection guidelines (please see Line 119), where  images with too low resolution will be excluded. For the resolution statistics, please refer to Table 9 in the Appendix, where we provide the detailed statistics for resolution on the z-axis.
>
> **8. Detailed information of senior radiologists**
>
> As introduced in Appendix B.2, the coarse masks generated by a pre-trained segmentation model are first refined by five junior radiologists with 5 years of experience in clinical scanning, and further supervised by three board-certified senior radiologists with 10 years of experiences.
>
>
> **9. How to convert DICOM to NIFTI**
>
> We utilize official tool from https://people.cas.sc.edu/rorden/mricron/dcm2nii.html for the conversion.
>
> **10. More details on ethical and responsible use of AMOS**
>
> As introduced in Appendix B.3, we adopt the CC BY-NC-SA (Attribution-NonCommercial-ShareAlike) license to constrain the data distributation, and ethical and responsible usage.
>
> **11. Clarification of the term "disease" used in Figure 3a**
>
> Thank you for the valuable comments. We have changed the title from "disease" to "disease/symptom" in the latest revision.
>
>
> **12. Missing citations**
>
> Thanks for the detailed review. We have added the citation [a] in the latest revision.
>
> **13. Rephrasing the claims to make it clear that AMOS is not the largest**
>
> Thanks for the suggestions which is helpful to improve the paper clarity. We have changed the claim to `making it the most comprehensive and diverse benchmark of its kind to date.`
>
> **14.  Clarifications on UNet and nnUNet**
>
> We are sorry for the misleading. We have added a further descriptation to clarify that both the two nets are also able to achieve multi-organ segmentation (see line 85).
>
> **15. Incoporating external training loss for small organs segmentation**
>
> It's a very good suggestion to introduce the carefully-designed trainig objectives to improve the segmentation performance. Since this papaer focuses on benchmarking numerous methods in a common-used training settings, we do not introduce the metioned losses at the current stage. However, we will add new experiments that incoporates the related losses for further analysis in the future.
>
> **16. Some Typos**
>
> Thanks for the detailed suggestions. We have fixed all the metioned typos in the latest revision (see the blue content in the main paper).
>
> [a] Antonelli, M., Reinke, A., Bakas, S., Farahani, K., Kopp-Schneider, A., Landman, B. A., ... & Cardoso, M. J. (2022). The medical segmentation decathlon. Nature Communications, 13(1), 1-13.

---

> > ### Comment · Reviewer_jTzG · 2022-08-29
> > **Thank you**
> >
> > Thank you for your responses. I have no further questions. I think this dataset will be of great value to the research community -- excellent work!

---

### Official Review · Reviewer_T6XQ · 2022-07-28
**A valuable and well-constructed dataset for the community and for benchmarking**

**Rating:** 8
**Confidence:** 4

**Strengths:**

Compared with other existing abdomen organ datasets, AMOS contains a larger number of scans. Further, AMOS provides abundant annotations of multi-modality, multi-center, multi-scanner, multi-phase, and multi-disease patients, covering 15 organs. Further, the dataset is comprehensively evaluated by different types of baseline methods, including the CNN-based methods, Transformer-based methods, and hybrid ones. This comprehensive experiment setting strongly proved the generalization capability of the dataset.

**Weaknesses:**

The quality of the dataset and paper content is excellent. More detailed information could be clarified in the paper.  For example, during the annotation review by humans, due to the decision making being subjective to each radiologist, is there any way to deal with the differences between different radiologists? Could the paper release the standards for decision making between different radiologists.


**Additional Feedback:**

The paper is very well written. Here is a minor remarks about typo:

In 3.1 Dataset Contribution → Data Overview

“Following the standard clinical acquisition protocols, … at SLonggang District 109 People’s Hospital and Longgang District People’s Hospital,”

It seems “S” incorrectly occurs here? Should it be “Longgang District” rather than “SLonggang District”?


**Clarity:**

Yes, the paper is well written and presents content in an organized way.


**Correctness:**

The dataset is constructed in a sound and correct way.


**Documentation:**

Yes, sufficient details and documentation has been provided.


**Relation To Prior Work:**

It is clearly discussed in the ‘Related Works’ section how this work relates to prior publicly available datasets and the previous methods for abdominal single- and multi-organ segmentation.

**Summary And Contributions:**

The paper proposed a large-scale abdominal organ segmentation dataset with transparent data collection, annotation, split, and distribution details. Furthermore, the data in AMOS are collected from multi-center, multi-vendor, multi-modality, multi-phase, multi-disease patients, each with voxel-level annotations of 15 abdominal organs. The dataset construction is transparent and reasonable enough. Further, the authors utilized different types of baseline methods for dataset evaluation.

---

> ### Author Response · Authors · 2022-08-24
> **Reply to Reviewer T6XQ**
>
> Dear reviewer T6XQ,
>
> Thank you for your comments. We answer the raised questions and concerns below. We have included all discussions and results in our revised manuscript (blue content).
>
>
> **1. Clarifications on more detailed information of the dataset**
>
> For the consistency of the annotation review, specifically, each senior physician will first individually review and record their comments, including a description of the problem and the corresponding image location, and then the comments will be aggregated and discussed to reach a final consensus opinion. We have included the detailed descriptions of the annotation reviewing process in the revised version.
>
> **2.Some typos**
>
> Thanks for your suggestiosn. We have fiexed all the typos in the revised manuscript. Please see the blue content in the updated version.

---

### Official Review · Reviewer_WL5C · 2022-07-28
**Very useful contribution to Medical Imaging Community**

**Rating:** 9
**Confidence:** 4

**Strengths:**

As stated earlier, the size of the dataset and the number of organs segmented opens avenue for designing and validating methods for multiple organ segmentation taking their relative position and shape/size into account. Multivendor scans with varied diseases conditions adds to the challenge of learning and helps in benchmarking robust algorithms.

Consultation with multiple senior radiologists when finalizing the segmentation boundary is a big plus. Many of the datasets that involve only a single domain expert, for example some mentioned in this paper, may inherit the bias of the expert, which may then be learnt/amplified by the learning algorithms.

Publicly available leader board as hosted in grand-challenge.org is useful in tracking the state-of-the-art.

**Weaknesses:**

Not necessarily a weakness, but the use of default parameters for training many of the baselines (no hyperparameter tuning), might hinder making a rigorous statement about the experiment results.

**Additional Feedback:**

Line 184 mentions Fig 4. But fig 4 is in supplementary text. Perhaps, it is better to indicate that the figure is in the relevant appendix section (not in the main text).

Section 3.3 Evaluation Metrics. Please mention at what tolerance level was the NSD calculated?

Line 248 What do you mean “we have also observed a significant gap between mDice and mNSD as previously mentioned by [15]?” Perhaps, concrete examples of the difference (for which organs, these metrics were significantly different?)  and interpretation of what that means.

 Line 226-227 “All models are optimized by the empirical risk minimization (ERM), which trains the model to minimize the average training loss.” This sentence may be redundant. It is a given.

### Typo and Grammar:

Line 25 Nerveless -> Nevertheless
Table 7 performacne -> performance

Line 144 Training data can **differs** the testing data -> training data can **differ from** the testing data

Line 157 Perhaps, Data Format is a better paragraph title here rather than Data Distribution.

Line 188 and its **closed datasets,** BTCV -> and its **closest dataset,** BTCN

Line 268 Since AMOS **owns** more modalities -> Since AMOS **contains** more modalities

Please check for any additional typos in the draft.

**Clarity:**

Yes, the major points come through clearly. There are few typos, redundancies, word choice and omissions.

**Correctness:**

Yes, the dataset is a very useful contribution in terms of number of organs segmented and the number of samples on which the full spectrum of organs is segmented.  Required caution has been made in making the dataset bias free and preserving anonimity.

**Documentation:**

Yes, the dataset is hosted as a challenge in grand-challenge.org. Documentation of details of data collection, intended use are clearly stated.

**Relation To Prior Work:**

Yes, relationship with various other public multi-organ, multi-modal and uni-modal datasets are discussed. Network architectures in the literature related to multi-organ(and partial organ) segmentation are also discussed.

**Summary And Contributions:**

The paper contributes a publicly available dataset of CT and MRI Image and segmentation pairs with 15 different organs segmented for all of the images. There are 100 MRI and 500 CT scans in total. This contributes to fulfil an important gap where various datasets of single organs are available individually but multiple organ segmentation are not available in the amount contributed by this paper. This helps in training data hungry deep learning algorithms for segmenting organ boundaries in abdomen region which can be useful for reducing manual segmentation effort.
Benchmarking of few representative baselines on segmentation tasks, transfer learning, cross-modality learning etc. are also done.

---

> ### Author Response · Authors · 2022-08-24
> **Reply to Reviewer WL5C**
>
> Dear reviewer WL5C,
> Thank you for appreciating our approach. We will address your comments below.
>
> **1. No hyperparameter tuning**
>
> Thanks to point out the weakness. In fact, we did a series of experiments with different hyper-parameters for Swin-Unetr [a]. However, it turned out that we found no significant differences among the experimental results. Thus, we finally adopted the default configuration for simplicity and fair comparisons. Nevertheless, we fully agree with your comments, and we will conduct more thorough experiments in the future to conduct further comparisons.
>
> **2. Improper reference of Figure 4**
>
> Thanks for your detailed suggestions. We have added the descriptions in Figure 4 for better clarity. Please check the latest revision.
>
> **3. Clarifications on the tolerance level of NSD**
>
> We use tolerance_mm = 1 for all organs.
>
> **4. Provide concrete examples of the difference between two metrics, and the interpretation of what that means.**
>
> As demonstrated in Table 10 in the appendix, the Dice scores showed significant differences with respect to the NSD scores in almost all organs. Specifically, the Dice similarity coefficient (DSC), a region-based measure, is used to evaluate the region overlap. Normalized surface Dice (NSD), a boundary-based measure, is used to evaluate how close the segmentation and ground truth surfaces are to each other at a specified tolerance. Even though both the two results show their effectiveness in evaluating the current abdominal segmentation algorithm, the accuracy of the boundary segmentation still needs further improvement.
>
> **5.Redundant expressions and typos**
>
> Thanks for the detailed comments and suggestions, which improves our paper for better clarity. We have fixed all the typos and redundant sentences in the revised manuscript. Please see the blue content in the updated version.
>
> [a] Hatamizadeh, A., Nath, V., Tang, Y., Yang, D., Roth, H. R., & Xu, D. (2022). Swin unetr: Swin transformers for semantic segmentation of brain tumors in mri images. In International MICCAI Brainlesion Workshop (pp. 272-284). Springer, Cham.

---

> > ### Comment · Reviewer_WL5C · 2022-08-27
> > **Satisfied with the update**
> >
> > I would like to thank the authors for updating the paper. I am satisfied with the update and addresses most of my inquiries.

---

### Official Review · Reviewer_3Rsc · 2022-07-28
**Abdominal Multi-Organ Dataset for Medical Image Segmentation**

**Rating:** 7
**Confidence:** 4

**Strengths:**

-The dataset presented by the authors contains the maximum annotations per scan as stated in *Table 1* , which is a significantly larger and at the same diverse as well (with reference to *Figure 4*) than the previous datasets, thus can serve a good benchmark in future

-The authors have meticulously conducted experiments across different state-of-the-art models

-The dataset presented by the authors shows potential in transfer learning and cross-modality learning a fair setup




**Weaknesses:**

* With reference to page number 6, Table 3, the authors left some important multi-organ dataset namely *AbdomentCT-1K* dataset for benchmarking, which leaves the comparison somewhat incomplete, does that mean a pre-trained model on *AbdomentCT-1K*
outperforms over pre-trained *AMOS* or the performance gain is marginal compared ?


* As stated on line number *257*:\
         *The results demonstrate that there is indeed a data distribution shift caused by collecting MRI samples from different scanners that
         hinders the model from performing consistently on two sub-sets*\
         The authors could have conducted a few experiments specifically comparing AMOS-MRI with other MRI datasets rather than stating
         Out of Distribution (OOD) as stated in Table 6. Further to tackle this ood, do authors  intend to add more data in future or have any plan to mitigate the OOD?




**Additional Feedback:**

With reference to line number 115-117

*"To meet this purpose, we cautiously select data, generated
116 by two institutes from 2018 to 2021, based on the following criteria: 1) Patients should be diagnosed
117 with abdominal tumors/abnormalities, while the ones with normal abdomen will be excluded"*

It would be great if you can include the data with normal abdomen to provide even more diversity and generalization for transfer learning tasks



**Clarity:**

The paper clearly explains the data acquisition and annotation procedure. It also highlights the vital dataset statistics and experiment setup to benchmark other SOTA models.

Overall well articulated.



**Correctness:**

Please correct the Appendix B.5 to Appendix B.4 throughout the paper

**Misc:**
\
W.r.t Appendix,On line number 573:
*normalize truncated voxels values by subtracting 50 and dividing by 141*

Any specific reason why did you chose these values for normalization ?

**Documentation:**

Overall the paper is clear and the authors provide enough implementation detail to reproduce the work, in my opinion.


**Ethics:**

The dataset abides to the ethical practices.

**Relation To Prior Work:**


The authors provide adequate context along with vital statistics among abdominal datasets to substantiate the difference across various parameters. Also, the authors have highlighted all the distinguishing features of this work in relation to prior art.



**Summary And Contributions:**

This paper presents AMOS a large scale abdominal multi-organ dataset from diverse clinical scenarios to serve as a benchmark to study robust segmentation algorithms. The dataset is comprehensively annotated with semi-automatic annotation workflow along with clinical experts to reduce the bias. Authors establish the diversity and scale of the dataset by comparing AMOS with existing abdominal datasets along the different parameters: organs, modality, scans, slices and annotations per scan. Further, few experiments were setup to benchmark several state-of-the-art medical segmentation models and evaluate them on existing abdominal datasets. The paper also focuses on out of distribution model generalization, transfer learning and cross modality learning, further substantiate it with experiment results. The authors also elucidate the data curation process and address its limitations.

---

> ### Author Response · Authors · 2022-08-24
> **Reply to Reviewer 3Rsc**
>
> Dear Reviewer 3Rsc,
> Thank you for the valuable comments and suggestions on our work. We will address your concerns below.
>
> **1. Incomplete comparsion of Abdmonetion-1K in Table 3**
>
> Perhaps a slight misunderstanding. Please note that models are trained on the source dataset and tested on the target dataset. Thus, there should be no overlap between the source dataset and the target dataset. However, since the chosen target dataset in our paper (e.g., MSD-Liver, MSD-Spleen, and MSD-Pancreas) is a sub-set of Abdmonetion-1K, we omit it for avoiding unfair comparison.
>
>
> **2. Clarifications on the extension of OOD experiments**
>
> Thanks for suggestion. We will compare AMOS-MRI with other datasets if they share enough organ annotations. For furture work, if funding permits, we will further enlarge the dataset by involving more diverse cases from more centers. In addition, we plan to release unlabeled data for both the source and target domains, with the hope to inpsire the development of corresponding methods.
>
> **3. Incorrect Appendix**
>
> Thanks for the suggestions. We have fixed it in the latest revision.
>
> **4. Clarifications on the normalization**
>
> Thanks, we followed the common practice of CT image preprocessing to perform normalization [a]. Specifically, we statistically counted the HU values of all foreground regions in the dataset and then selected the mean and standard values for normalization.
>
> **5. Including the normal cases**
>
> Thanks for this suggestion, please see **Clarifications on including more normal cases** in the **Author Response to All Reviewers**.
>
> [a] Isensee, F., Jaeger, P. F., Kohl, S. A., Petersen, J., & Maier-Hein, K. H. (2021). nnU-Net: a self-configuring method for deep learning-based biomedical image segmentation. Nature methods, 18(2), 203-211.

---

> > ### Comment · Reviewer_3Rsc · 2022-08-29
> > **Satisfied**
> >
> > I would like to thank the authors for addressing my concerns. Overall I am fine with the explanations provided

---

### Official Review · Reviewer_1Y16 · 2022-07-29
**AMOS dataset for abdominal segmentation from CT and MRI scans.**

**Rating:** 9
**Confidence:** 5
**Clarity:** Yes, the paper is well written.

**Strengths:**

- A large, diverse dataset for a crucially important CAD problem in medical imaging.
- Clear experiments and comparisons explain AMOS's benefits over the existing open source abdominal datasets.
- they provide clear use cases for their dataset and how they envision the community can use it.
-they provide strong baselines and plan to release the code making this dataset easier to use. They have posed this dataset as a MICCAI challenge, which will also help spread the use in the community.

**Weaknesses:**

The authors have captured the restrictions of this work pretty well.
- As mentioned in the paper, it is a subset of the Asian population with data from two hospital systems with similar protocols for multiple scanners.


**Additional Feedback:**

Minor comments:
- the authors decided not to include normals and focus only on diseased cases. Having normals would also have been beneficial, but that is in no way discounting the value of this dataset.
- the minor typos in the paper, which a good revision can quickly fix.
- On lines 188-189: The authors mention they pre-trained the models between AMOS and BTCV and tested them on the others. Did they further fine-tune the algorithm using the training set of the other datasets or use the pre-trained models as is? Can you clarify?
- On lines 322-323: Can the authors clarify what they mean by solving privacy and social effects by obtaining the research ethics committee approval? It is unclear what social and privacy issues the committee helped solve.

**Correctness:**

Yes, the claims in the submission are correct and sound with reasonable baselines and benchmarks.

**Documentation:**

Yes, there is sufficient detail on the dataset creation, maintenance, and use cases.

**Ethics:**

No.

**Relation To Prior Work:**

Yes, the paper has good evaluations on comparison with other datasets and motivation for creating this dataset.

**Summary And Contributions:**

The authors present a large, diverse dataset of abdominal CT and MRI scans with pixel-level annotations of 15 organs. The paper comprehensively explains the data collection procedures and the dataset's limitations, present their use cases, and provides baseline experiments for the community to build upon.

Overall this is an excellent resource for further developing CAD for multi-organ abdominal segmentation.

---

> ### Author Response · Authors · 2022-08-24
> **Reply to Reviewer 1Y16**
>
> Dear Reviewer 1Y16,
>
> Thank you for your comments. We answer the raised questions and concerns below. We have included all discussions and results in our revised manuscript (blue content).
>
> **1. Restrictions of AMOS**
>
> Thanks for pointing out the restrictions on the socio-demographic features. We kindly admit that it is a limitation since only the Asian population from two hospitals is included in our dataset.  For future work, if funding permits, we will further enlarge the dataset by involving more diverse patients from more centers.
>
> **2. Including normal patients would be better**
>
> Thanks for this suggestion, please see **Clarifications on including more normal cases** in the **Author Response to All Reviewers**.
>
> **3. Some typos**
>
> We have fixed all the typos in the revised manuscript. Please see the blue content in the updated version.
>
>
> **4. Clarifications on the fine-tuning settings**
>
> We do not fine-tune with the pre-trained checkpoints. Instead, we directly load the pre-trained weights and evaluate them on the other datasets. The experimental purpose is to show the superior data capacity and quality of our AMOS. That is to say, even without the fine-tuning processes, the pre-trained models on AMOS could achieve considerable performances.
>
> **5. Solutions to social and privacy issues**
>
> We follow the guidance from the hospitals to collect the data. The privacy and social effects are carefully addressed with the help of professional radiologists. The research ethics committee conduct a further approval check on the privacy and social issues.

---

> > ### Comment · Reviewer_1Y16 · 2022-08-29
> > **My comments have been addressed.**
> >
> > I thank the authors for addressing my comments. As such, I don't have any additional comments.

---

### Author Response · Authors · 2022-08-24
**Author Response to All Reviewers**

Dear all reviewers,

We sincerely appreciate all reviewers for their time and efforts in reviewing our paper. We are glad to find that reviewers recognized the following contributions of our work:

- 1. **Unique contributations of AMOS to the research community.** We sincerely appreciate that the unique contributations have been recognized by all the reviewers, including the large scale [1Y16,3Rsc,WL5C,T6XQ,jTzG,vCQ1,CcuT], high diversities[1Y16,3Rsc,WL5C,T6XQ,jTzG,vCQ1,CcuT], multiple functions [1Y16,3Rsc,vCQ1] and so on.

- 2. **Sufficient experiments and comparisons.** We are glad to find that most reviewers have reached consensuses that our experiments are sufficient [1Y16,3Rsc,T6XQ,jTzG,CcuT] and comparisons clearly explain AMOS's benefits over the existing open source abdominal datasets [1Y16,3Rsc,T6XQ,jTzG,CcuT].

- 3. **Clear potential usuages of AMOS.** The potential usuages of AMOS have been acknowledged by reviewers [1Y16,3Rsc,WL5C,T6XQ,jTzG,vCQ1,CcuT], including the cross-modality learning, transfer Learning and so on.

- 4. **Well-written and clearly-presented paper.** We also apperiacate the clear organization of the paper has been acknowledged by the reviewers [1Y16,3Rsc,WL5C,T6XQ,jTzG,vCQ1,U7Pt,CcuT].

We also thank all reviewers for their insightful and constructive suggestions, which help further improve our paper. In addition to the pointwise responses below, we summarize the major revision in the rebuttal according to the reviewers’ suggestions.

- 1. **Appendix updating.[3Rsc,jTzG,vCQ1,U7Pt]** We are sorry that we have appended an uncomplete and non-latest appendix to the main paper when we submmitted. In the next revision, we have updated the appendix to the latest version to make sure that there's no missing and uncomplete sub-sections.
- 2. **Clarifications on including more normal cases. [1Y16,3Rsc,vCQ1]**  In fact, including the normal cases will be beneficial to general organ segmentations. However, since AMOS focuses clinical settings, where the abnormal cases are more crucial for practical segmentation, we do not include the normal organs at the current stage. For future work, if funding permits, the normal cases will be considered to be included for dataset diversities.

We hope our pointwise responses below could clarify all reviewers’ confusion and alleviate all concerns. We thank all reviewers’ efforts and time again.

Best regards,
Authors

---

### Meta-Review · Area_Chair_sJYt · 2022-08-31

**Recommendation:** Accept
**Confidence:** 5

**Metareview:**

The authors have done a good job in addressing the review comments and all the 8 reviewers are positively in support of accepting this work. I would like to reiterate to authors to please check if their data collection follows standard ethical guidelines of medical data collection and release One example of guideline can be - EU guideline - https://ec.europa.eu/info/funding-tenders/opportunities/docs/2021-2027/horizon/guidance/ethics-and-data-protection_he_en.pdf

---

### Decision · Program_Chairs · 2022-09-16

Accept